# Cardiac Magnetic Resonance Shows Improved Outcomes in Patients with an ST-Segment Elevation Myocardial Infarction and a High Thrombus Burden Treated with Adjuvant Aspiration Thrombectomy

**DOI:** 10.3390/jcm11175000

**Published:** 2022-08-25

**Authors:** Wojciech Zajdel, Tomasz Miszalski-Jamka, Jarosław Zalewski, Jacek Legutko, Krzysztof Żmudka, Elżbieta Paszek

**Affiliations:** 1Clinical Department of Interventional Cardiology, The John Paul II Hospital, Pradnicka 80, 31-202 Krakow, Poland; 2Department of Radiology and Imaging Diagnostics, The John Paul II Hospital, Pradnicka 80, 31-202 Krakow, Poland; 3Department of Coronary Artery Disease and Heart Failure, Jagiellonian University Medical College, sw. Anny 12, 31-008 Krakow, Poland; 4Clinical Department of Coronary Disease and Heart Failure, The John Paul II Hospital, Pradnicka 80, 31-202 Krakow, Poland; 5Department of Interventional Cardiology, Institute of Cardiology, Jagiellonian University Medical College, sw. Anny 12, 31-008 Krakow, Poland; 6Department of Thromboembolic Disorders, Institute of Cardiology, Jagiellonian University Medical College, sw. Anny 12, 31-008 Krakow, Poland

**Keywords:** aspiration thrombectomy, cardiac magnetic resonance, left ventricular remodelling, microvascular obstruction, ST-segment-elevation myocardial infarction

## Abstract

There is a discrepancy between epicardial vessel patency and microcirculation perfusion in a third of patients treated with percutaneous coronary intervention (PCI) for ST-segment elevation myocardial infarction (STEMI). Optimization with aspiration thrombectomy (AT) may reduce distal embolization and microvascular obstruction. The effect of AT in the treatment of STEMI is debatable. The purpose of this study was to use cardiac magnetic resonance (CMR) to determine whether AT influences microvascular obstruction (MVO), infarct size and left ventricular (LV) remodelling in STEMI patients. Sixty STEMI patients with a thrombus-occluded coronary artery were randomized in a 2:1 fashion to receive PCI proceeded by AT (AT + PCI group), or PCI only. MVO, myocardial infarct size and LV remodelling were assessed by CMR during the index hospitalization and 6 months thereafter. The majority of patients had a large thrombus burden (TIMI thrombus grade 5 in over 70% of patients). PCI and AT were effective in all cases. There were no periprocedural strokes. CMR showed that the addition of AT to standard PCI was associated with lesser MVO when indexed to the infarct size and larger infarct size reduction. There were less patients with left ventricle remodelling in the AT + PCI vs. the PCI only group. To conclude, in STEMI patients with a high thrombus burden, AT added to PCI is effective in reducing infarct size, MVO and LV remodelling.

## 1. Introduction

The ST-segment elevation myocardial infarction (STEMI) annual incidence rate is estimated at three million cases worldwide [1,2]. Despite dynamic developments in the diagnosis and treatment of myocardial infarction, the morbidity and mortality rates in survivors remain high [3]. Percutaneous coronary intervention (PCI) is fundamental in the treatment strategy of STEMI patients [4]. Restoration of the infarct-related artery (IRA) patency, at least theoretically, enables coronary microcirculation salvage and prevents myocardial necrosis. However, modern imaging methods have revealed a discrepancy between epicardial vessel patency and microcirculation perfusion in at least one third of STEMI patients treated with PCI [5,6]. 

Coronary microvascular obstruction (MVO), which is associated with the no-reflow phenomenon, is partly caused by distal embolization [7,8]. A recent study showed that enhanced thrombin generation in patients with STEMI is associated with the extent of necrosis as assessed by CMR [9]. It is debatable to what extent MVO impacts myocardial necrosis and left ventricle (LV) remodelling [10,11]. Therefore, the concept of aspiration thrombectomy (AT) as a mechanical adjuvant intervention during PCI for thrombus removal, possibly resulting in a reduced distal embolization, is worth investigating. 

In recent years, novel advanced imaging techniques of the coronary arteries and myocardial tissue, including highly sophisticated cardiac magnetic resonance (CMR) modalities, became available. This allowed for a precise assessment of the impact of thrombus burden reduction on infarct size, scarring and LV remodelling [12,13]. Although echocardiography is a faster, cheaper and easily accessible method of left ventricle function assessment, CMR is a significantly more precise and sensitive tool, enabling the assessment of parameters such as MVO or scar retraction which are unavailable in echocardiography [14]. 

Therefore, in the present study we aimed to evaluate whether the use of AT as an adjuvant therapy to standard PCI in STEMI patients is associated with MVO, infarct size and left ventricular (LV) remodelling in STEMI patients using CMR perfusion assessment.

## 2. Materials and Methods

### 2.1. Study Population

We included 60 consecutive patients diagnosed with STEMI, who had a total occlusion of the infarct related artery (TIMI Flow 0–1) with a thrombus visible on the coronary angiogram and who were referred for PCI. The exclusion criteria were as follows: a symptom onset > 12 h prior to admission, left bundle branch block, haemodynamic instability (including cardiogenic shock, pulmonary oedema, NYHA III or IV class symptoms), a prior STEMI in the same location, any contraindication for glycoprotein IIb/IIIa inhibitor (GPI) administration, extreme tortuosity of the IRA, a reference diameter < 2.5 mm proximal to IRA occlusion, and any contraindication for CMR. 

### 2.2. Study Protocol

The study protocol was summarized in Figure 1. 

Directly after coronary angiography, patients were randomly assigned in a 2:1 ratio to one of two treatment groups: 

Group I—the AT plus PCI group (AT + PCI group) where thrombus aspiration was performed, followed by routine PCI with stent implantation

Group II—where PCI was performed, without AT (PCI only group).

All patients received identical standard premedication treatment periprocedurally and during the follow-up period, in adherence to the European Society of Cardiology Guidelines [4]. All patients had received a loading dose of acetylsalicylic acid (ASA, 300 mg p.o.) and clopidogrel (600 mg p.o.) before the procedure. Newer P2Y12 antagonists were unavailable at the time the study was performed. Intravenous, unfractionated heparin (UFH) was administered in all patients with a target activated clotting time (ACT) of 200–350 s. In selected cases, at the operator’s discretion, intravenous Abciximab was added in a body weight adjusted dose. 

In the AT + PCI group, AT was performed using the mechanical Rescue System™ (Boston Scientific Corp., Marlborough, MA, USA) with an active lumen diameter of 0.042 inches. Aspiration was performed from the end of the guiding catheter to the site of the occlusion and distally, with at least two passages or to maximum elimination of the visible thrombus. If an attempt to perform AT across the lesion was unsuccessful, a predilation with a small balloon (≤2.0 mm) was performed and aspiration was attempted again. In all cases, following the opening of the IRA, the procedure was optimized by a stent implantation with a high-pressure technique. Post procedural distal embolization and TIMI Myocardial Perfusion Grade (TMPG) were assessed. Angiographic analysis was performed post hoc using Acom PC (Siemens, Munich, Germany) by two independent, experienced operators. 

The primary endpoints were: the extent of MVO, MI size and the occurrence of LVEDV enlargement (∆LVEDV, defined as an increase in LVEDV of 12% or more) assessed by CMR [15]. Demographic and biochemical variables, and atherosclerosis risk factors such as arterial hypertension, dyslipidaemia, diabetes mellitus, smoking, and 12-lead standard ECG were recorded in all patients. 

### 2.3. CKMB Release

Blood levels of creatinine kinase MB isoform (CKMB) were evaluated on admission, 24 and 48 h after reperfusion. The sum of CKMB released was calculated as the area under the release curve (AUC, U × h) in order to estimate infarct size, as part of an established method [16,17].

### 2.4. Resolution of ST-Segment Elevation on 12-Lead ECG

Standard electrocardiograms (25 mm per second recording speed and 1 mV per centimetre voltage) were recorded for all patients on admission, directly post-procedurally, one and two hours after reperfusion. The sum of ST segment elevation in the J point was measured by two independent, experienced cardiologists in accordance with the method previously described [18].

### 2.5. CMR Acquisition 

The primary and secondary endpoints were evaluated using CMR (1,5T Magnetom Sonata Maestro Class, Siemens, Munich, Germany). CMR was performed twice: between day 3 and 5 of the index hospitalization (CMRindex) and 6 months post PCI (CMRfollow-up). CMRindex encompassed cine, morphologic and late gadolinium enhancement (LGE) imaging, whereas CMRfollow-up only encompassed cine and LGE imaging. Cine imaging was performed in apical left ventricular long and short-axis views encompassing the entire LV myocardium with balanced steady-state–free precession gradient echo (slice/gap thickness 8/0 mm, matrix = 256 × 192, in-plane resolution = 1.4 × 1.4 mm^2^, TR = 3.0 ms, TE = 1.3 or 1.2 ms, flip angle = 59 or 700). Morphologic images were acquired in short axis views with T2-weighted triple inversion recovery (TIRM) (slice/gap thickness = 8/0 mm, matrix = 256 × 192, in-plane resolution = 1.4 mm × 1.4 mm, TR depending upon RR interval, TE = 65 ms, flip angle = 1800) and T1-weighted turbo spin echo imaging (slice/gap thickness = 8/0 mm, matrix = 256 × 192, in-plane resolution = 1.4 mm × 1.4 mm, TR depending upon RR interval, TE = 6.9 ms, flip angle = 1800). Ten minutes after intravenous infusion of 0.1 mmol/kg body-weight gadobutrol (Gadovist, Bayer Schering Pharma, Berlin, Germany) LGE imaging was performed in apical LV long axis and short-axis views with T1-weighted, segmented, inversion-recovery pulse sequence (slice thickness = 8 mm, gap thickness = 0 mm, matrix = 256 × 192, in-plane resolution = 1.4 × 1.4 mm^2^, TR = 650 or 700 ms, TE = 4.9 or 3.3 ms, flip angle = 30, TI to null normal myocardium).

### 2.6. CMR Analysis

The analysis was performed off-line using dedicated software (MASS Medis, Leiden, The Netherlands) by an independent reader (TMJ) blinded to clinical and angiographic data. As previously described, endocardial and epicardial LV borders were outlined on short-axis cine images to calculate ejection fraction (LVEF) as well as end-diastolic volume (LVEDV), and myocardial mass (LVmass) which was indexed to body surface area (LVMass index). The cut-off value for the occurrence of adverse LV remodelling six months after STEMI was defined as an enlargement of LVEDV of at least 12% [15]. TIRM images were quantitatively evaluated for the presence and location of hyperintense myocardial areas when compared to normal appearing myocardium. The cut-off value to delineate the hyperintense areas defined as areas at risk (AAR) was two standard deviations above the mean signal intensity of normal appearing myocardium. The volume of AAR was calculated and indexed to LV myocardial volume (AARindex). Based on quantitative evaluation, LGE images were assessed for the presence and location of hyperintense lesions in contrast to normal appearing myocardium (MI size). The cut-off value to delineate LGE lesion was 5 standard deviations above the mean signal intensity of normal appearing myocardium. The hypointense areas within LGE were defined as microvascular obstruction (MVO) lesions and were included in the area of LGE lesions. The size of MI was calculated as the volume of LGE and indexed to LV myocardial volume (MIindex). Similarly, the volume of MVO was quantified and expressed as the percentage of the volume of LGE lesion (MVOindex). To evaluate the effectiveness of the treatment, myocardial salvage index at CMRindex as an indicator of treatment effectiveness was assessed using the following formula: Myocardial Salvage Index (MSI) = (AAR vol − MI vol)/(AAR vol)

### 2.7. Statistical Analyses

Descriptive statistics were used to summarize the data of the cohorts. The distribution of data was evaluated using the Shapiro-Wilk test. Due to the lack of normal distribution, the results were shown as median [interquartile range, IQR]. Comparative statistics used the non-parametric U Mann–Whitney test for numerical variables and the chi-squared test for categorical variables. An essential level of significance was assumed at *p* = 0.05. Statistics were calculated using JASP 0.11.13. 

## 3. Results

### 3.1. Patients Characteristics

A total of 60 patients, predominantly men, were included into the study that comprised the AT + PCI Group (*n* = 40) and the PCI alone Group (*n* = 20). The median age was 55 [16.5] years in the AT + PCI group and 57 [20.0] years in the PCI only group, *p* = 1.0. There were no significant differences between the two groups in terms of atherosclerosis risk factors or comorbidities (Table 1). 

Additionally, both groups were homogenous with regard to baseline clinical and haemodynamic parameters, total ischemia time and angiographical parameters (Table 2). 

### 3.2. Angiography and PCI

The majority of patients presented with a high thrombus burden (TIMI Thrombus Grade 5 in 82.9% in the AT + PCI group, and 76.5% in the PCI only group, *p* = 0.40, Table 2). PCI was effective in all cases and thrombotic material was obtained in all AT + PCI patients. In a visual assessment, AT led to a significant thrombus reduction in all cases. The final, postprocedural TMPG and angiographically assessed distal embolization rates were similar between the groups (Table 2). 

The addition of AT to PCI enabled more frequent direct stenting in the AT + PCI group vs. the PCI only group (42.9% vs. 5.9%, *p* = 0.014, Table 2). Importantly, AT-assisted PCI was safe; there were no serious periprocedural complications, including stroke.

### 3.3. Enzymatic and ECG Assessment

There were no significant differences in the infarct size measured as AUC of CK-MB (10,787 U/L for the AT + PCI group vs. 6337 U/L for the PCI only group; *p* = 0.09; Table 3). The ST-segment resolution on ECG between the two groups was also similar in both timepoints (Table 3).

### 3.4. CMR Assessment

Of the 60 patients who were initially included, an optimal quality CMR was obtained in 52 patients (Figure 1). During the six-month follow-up period, two patients died, and one patient underwent an urgent surgery, precluding a repeated CMR assessment. Of the 49 follow-up CMRs performed, four could not be evaluated due to insufficient quality, leaving a final number of 45 patients with an index and follow-up CMR (28 in the AT + PCI group and 17 in the PCI only group).

The MVO/MI index was significantly lower in the AT + PCI group in comparison with the PCI only group (11.7 [IQR 9%] vs. 22.2 [IQR 26.9%], *p* = 0.009; Table 3). We observed that MI size reduction after 6 months was slightly larger in the AT + PCI group compared with that of the PCI only group (−7.8 [IQR 10.8] g vs. −4.5 [IQR 4.8] g; *p* = 0.03, Table 3). Moreover, patients treated with AT had a significantly lower increase in LVEDV during follow-up than patients who did not (median, 8.7 [IQR 19.3] mL vs. 20.0 [IQR 20.5] mL, *p* = 0.004, Table 3). Consistently, six months post index procedure, significantly fewer patients in the AT + PCI group exhibited adverse LV remodelling (∆LVEDV > 12%: 8.6% vs. 52.9%, *p* = 0.019; Table 3). Example CMR images of a patient with AT + PCI and PCI only are shown in Figure 2 and Figure 3, respectively.

## 4. Discussion

The most important findings from the present study are that the addition of AT to standard primary PCI in STEMI patients with a high thrombus burden results in a significantly smaller degree of MVO, a larger reduction of MI size and less LV remodelling at six months following PCI for STEMI, as assessed by CMR. 

This is particularly important, as our cohort represents a group of relatively young patients with a median age of 55 (16.5) years in the AT + PCI group and 57 (20.0) years in the PCI only group, with a high prevalence of anterior STEMI, in whom both the quality of life, and life expectancy due to the risk of adverse cardiovascular events should be particularly addressed. Reports indicate that in young STEMI patients, severe thrombus formation is more frequent when compared with older patients, and the left anterior descending artery is the predominant IRA that requires more complex management [19]. In a study by Zasada et al., thrombus formation in the IRA was diagnosed more frequently in patients below 40 years old [19]. In line with this, AT was performed twice as often in patients younger than 40 years old, as compared with older patients (15.5% vs. 7.3%, *p* < 0.001).

In addition, in the present study most patients presented with a high thrombus burden (more than 70% of patients with TIMI Thrombus Grade 5, Table 2), which is known to carry an increased risk of distal embolization by fragmented thrombus debris [20]. This results in an increase of infarct size, which is dependent on MVO [21]. Of note, thrombotic debris constitutes a biologically active material that may exacerbate local injury via inflammatory reactions [22]. Unfortunately, although many cardioprotective strategies against myocardial injury have been proposed, none has shown a clinically significant improvement in STEMI patients [21]. Some promise may be associated with an intracoronary oxygen therapy [23], or modulation of the microRNAs expression [24]. An association between expression levels of miR-1–3p and thrombotic plaques, as a potential prognostic factor for future cardiovascular events has been reported [25]. 

In the light of the above, it seems reasonable that removing even a portion of the obstructive material in the setting of a high thrombus burden should improve flow in the IRA and muscle viability. In fact, mechanical thrombectomy is gaining significance in the treatment of acute ischemic stroke, a disease with a similar pathophysiology to myocardial infarction, as it is caused by a thrombus occlusion of intracerebral arteries [26]. However, current guidelines do not recommend the routine use of AT in the treatment of STEMI [4]. This is partially due to findings from older studies, using CK-MB release and ST-segment resolution to assess the infarct size and adverse LV remodelling in relation to AT [17,27]. In the present study we also found a similar ST segment resolution and CKMB release in both study groups, which is consistent with previous findings. It must be emphasized that these parameters are suitable only for an estimation of infarct size, and are unable to distinguish between subtle differences on tissue level [17,27]. On the contrary, advanced CMR protocols have enabled a precise assessment of coronary microvasculature and the interstitium by utilizing contrast medium diffusion during the first pass, together with the microvascular perfusion resistance index [28]. Furthermore, CMR has proven its utility in detecting inflammation in the myocardial tissue [29]. 

Very few studies have evaluated the effects of AT on CMR parameters in STEMI patients and these have brought conflicting results [30,31,32]. Our CMR analysis shows that AT + PCI significantly reduced MVO, infarct size and prevented adverse LV remodelling. This finding is consistent with some previous reports [30,33]. LV remodelling after STEMI is an important indicator of the risk of heart failure and its consequences, including malignant arrythmias and cardiac death [34,35]. Still, the majority of reports on the subject showed no effect of routine AT on hard clinical endpoints [4]. Possibly, the subtle differences observed in the CMR may not immediately translate into adverse clinical outcomes, such as mortality, however LV remodelling and MVO contribute to the risk of heart failure development [10]. 

Interestingly, we found that infarct size was significantly lower when using AT. Similarly, a CMR substudy of 75 patients from the EXPIRA trial showed that AT reduced MI size three months post index procedure, and this translated to a lower mortality nine months post the initial STEMI hospitalization [36]. Also, in a CMR study by Zia et al. involving 60 STEMI patients, AT during primary PCI was associated with reduced myocardial oedema, myocardial haemorrhage, left ventricular remodelling and incidence of MVO [33]. 

Some safety concerns considering AT arose after the TOTAL trial which showed a slight increase in the incidence of ischemic stroke, following AT-assisted primary PCI in STEMI (0.3% vs. 0.1% in the first 48 h post index procedure) [37]. The fact that none of the AT-treated patients in our cohort experienced a stroke deserves a comment. It needs to be emphasized that AT is a multistep, complicated procedure, which may result in peripheral embolization. It is, however, a reliable tool in the hands of an experienced operator, provided that all safety measures are kept, most importantly a deep intubation of the coronary artery and maintaining a negative pressure throughout the whole procedure, up to a complete removal of the device.

Study limitations. This was a relatively small study, however with regard to group characteristics, ours is a representative of a typical STEMI population. Despite the small sample size, to our knowledge, this is the most complete evaluation of the effect of AT on CMR outcomes in PCI-treated STEMI patients. A follow-up CMR assessment was possible in 45 out of the initial 60 patients. Other authors report similar difficulties, with a 75% effective CMR follow-up [38,39]. Therefore, our conclusions must be treated with caution. They are primarily meant to emphasize that there still is need for further research into the role of AT in STEMI, especially in the influence it may have on more subtle endpoints rather than major adverse clinical events. Moreover, an additional assessment of a broader range of inflammatory and necrotic biomarkers could shed new light on the role of AT in the treatment of STEMI patients with a high thrombus burden. 

## 5. Conclusions

We found that in relatively young STEMI patients presenting with a large thrombus burden, AT adjuvant to primary PCI was associated with a reduction in microvascular obstruction and a decrease in MI size. Aspiration thrombectomy added to PCI significantly reduced the LVEDV increase, leading to an overall lower incidence of LV remodelling in comparison with patients treated with basic PCI. Taken together, our current findings suggest that adjunctive AT may benefit STEMI patients who are at high risk of distal embolization due to a large thrombus burden. 

## Figures and Tables

**Figure 1 jcm-11-05000-f001:**
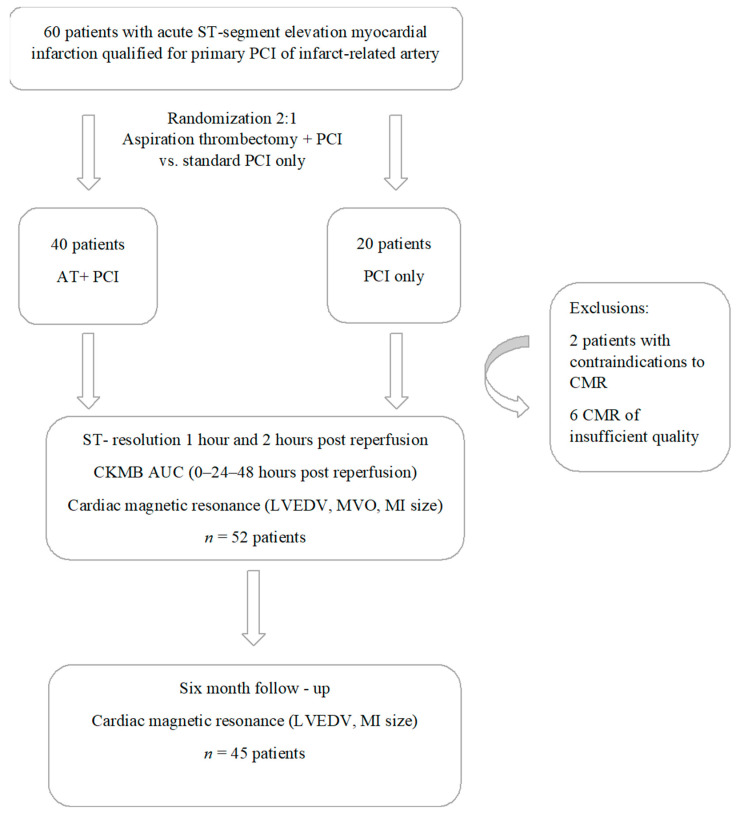
Flowchart detailing the study protocol. Abbreviations: AT—aspirational thrombectomy; AUC—area under the curve; CKMB—creatinine kinase MB isoform; CMR—cardiac magnetic resonance; LV—left ventricle; LVEDV—left ventricle end—diastolic volume; LVEF—left ventricle ejection fraction; MI—myocardial infarction; MVO—microvascular obstruction; PCI—percutaneous coronary intervention.

**Figure 2 jcm-11-05000-f002:**
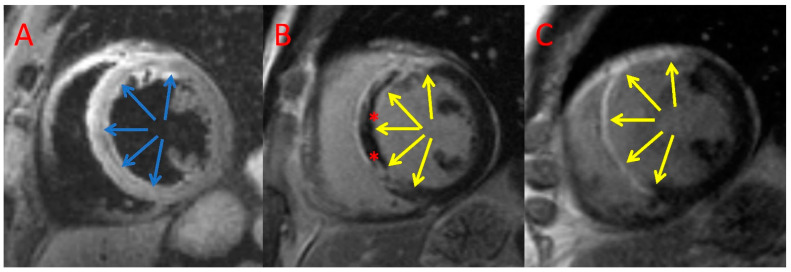
Example CMR images of a patient treated with PCI only. Index T2-weighted triple inversion recovery (**A**) and late gadolinium enhancement imaging (**B**) and CMR follow-up late gadolinium enhancement (**C**) imaging acquired in the short axis. Blue arrows—area at risk; yellow arrows—late gadolinium enhancement; *—microvascular obstruction.

**Figure 3 jcm-11-05000-f003:**
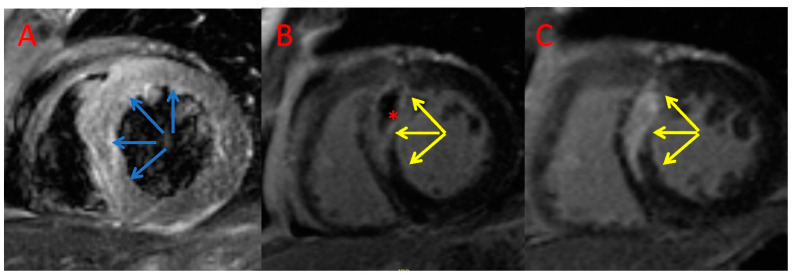
Example CMR images of a patient treated with an AT + PCI. Index T2-weighted triple inversion recovery (**A**) and late gadolinium enhancement imaging (**B**) and CMR follow-up late gadolinium enhancement (**C**) imaging acquired in the short axis. Blue arrows—area at risk; yellow arrows—late gadolinium enhancement; *—microvascular obstruction.

**Table 1 jcm-11-05000-t001:** Baseline patient characteristics.

	AT + PCI (*n* = 35)	PCI Only (*n* = 17)	*p* Value
**Age (years)**	55 (16.5)	57 (20.0)	1.00
**Body mass index (kg/m^2^)**	27.8 (3.4)	28.4 (3.9)	0.67
**Sex: male (*n*, %)**	23 (65.7%)	14 (82.4%)	0.80
**Hypertension (*n*, %)**	26 (74.3%)	6 (35.3%)	0.23
**Diabetes (*n*, %)**	8 (22.9%)	3 (17.6%)	0.43
**Dyslipidaemia (*n*, %)**	26 (74.3%)	12 (70.6%)	0.42
**Smoking (*n*, %)**	23 (65.7%)	8 (47.1%)	0.30
**Parameters on admission**			
Total ischemia time (min)	240 (110)	300 (250)	0.37
Systolic blood pressure (mmHg)	125 (32)	130 (30)	0.55
Heart rate (beats per minute)	80 (15)	75 (30)	0.96
Killip Class 0–1 (*n*, %)	34 (97.1%)	16 (94.1%)	0.26
Previous myocardial infarction (*n*, %)	3 (8.6%)	1 (5.9%)	0.85
Anterior myocardial infarction (*n*, %)	14 (40.0%)	10 (58.8%)	0.37

Results were shown as median (interquartile range) for continuous variables or *n* (%) for categorical variables. Abbreviations: AT—aspirational thrombectomy; PCI—percutaneous coronary intervention.

**Table 2 jcm-11-05000-t002:** Angiographic parameters.

	AT + PCI (*n* = 35)	PCI Only (*n* = 17)	*p* Value
**Multivessel disease (*n*, %)**	23 (65.7%)	12 (70.6%)	0.60
**IRA reference diameter (mm)**	3.12 (0.38)	3.02 (0.56)	0.40
**IRA location (*n*, %)**			0.41
LAD	16 (45.7%)	10 (58.8%)	
LCx	3 (8.6%)	0 (0%)	
RCA	16 (45.7%)	7 (41.2%)	
**Rentrop Grade (*n*, %)**			0.08
2	10 (28.6%)	3 (17.6%)	
1	14 (40.0%)	3 (17.6%)	
0	11 (31.4%)	11 (64.7%)	
**TIMI Thrombus Grade (*n*, %)**			0.40
5	29 (82.9%)	13 (76.5%)	
4	5 (14.2%)	2 (11.8%)	
3 or less	1 (2.6%)	2 (11.8%)	
**Final TMPG (*n*, %)**			0.26
2 or more	30 (85.7%)	10 (58.8%)	
1 or less	5 (14.3%)	7 (41.2%)	
**Embolization (*n*, %)**	9 (25.7%)	4 (23.5%)	0.86
**Direct stenting (*n*, %)**	15 (42.9%)	1 (5.9%)	0.014

Results were shown as median (interquartile range) for continuous variables or *n* (%) for categorical variables. Abbreviations: AT—aspirational thrombectomy; IRA—infarct-related artery; TIMI—thrombolysis in myocardial infarction; TMPG—TIMI Myocardial Perfusion Grade; PCI—percutaneous coronary intervention.

**Table 3 jcm-11-05000-t003:** Biochemical, electrocardiography and cardiac magnetic resonance parameters.

	AT + PCI (*n* = 28)	PCI Only (*n* = 17)	*p* Value
**CKMB AUC (U/l)**	10,787 (12,815)	6337 (5597)	0.09
**ST segment elevation**	18 (12.0)	15 (10.0)	0.32
**on admission**			
**ST segment elevation post PCI**			
Directly post PCI	7 (5.8)	7 (8.0)	0.47
1 h post PCI	4 (4.5)	4 (4.0)	0.72
2 h post PCI	3 (3.0)	3 (3.0)	0.27
**LVEDV (mL)**			
Index	142.6 (40.1)	127.8 (45.5)	0.54
Follow-up	149.2 (54.8)	158.0 (75.0)	0.26
∆LVEDV	8.7 (19.3)	20.0 (20.5)	0.004
**LV remodelling**			
∆LVEDV >12% (*n*, %)	3 (8.6%)	9 (52.9%)	0.019
**LVEF (%)**			
Index	37.8 (16.7)	45.7 (14.7)	0.96
Follow-up	44.2 (19.4)	43.5 (11.4)	0.98
∆LVEF	5.8 (10.1)	4.0 (6.3)	0.41
**MI size (g)**			
Index	28.0 (21.5)	25.0 (15.0)	0.39
Follow-up	19.8 (13.0)	16.9 (12.2)	0.55
MI size reduction	−7.8 (10.8)	−4.5 (4.8)	0.03
**MI size/LV mass (%)**			
Index	18.7 (11.3)	15.8 (8.5)	0.23
Follow-up	14.0 (11.7)	15.4 (9.8)	0.89
**MVO (g)**	4.0 (5.4)	6.0 (6.1)	0.26
**MVO/MI (%)**	11.7 (9.0)	22.2 (26.9)	0.009
**MSI (%)**			
Index	0.5 (0.3)	0.4 (0.3)	0.42
Follow-up	0.7 (0.2)	0.6 (0.2)	0.11

Results are shown as median (interquartile range) for continuous variables or *n* (%) for categorical variables. Abbreviations: AT—aspirational thrombectomy; CKMB—creatinine kinase MB isoform; LV—left ventricle; LVEDV—left ventricle end—diastolic volume; LVEF—left ventricle ejection fraction; MI—myocardial infarction; MVO—microvascular obstruction; PCI—percutaneous coronary intervention.

## Data Availability

Not applicable.

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
