# Peer review of "Cardiac Magnetic Resonance Shows Improved Outcomes in Patients with an ST-Segment Elevation Myocardial Infarction and a High Thrombus Burden Treated with Adjuvant Aspiration Thrombectomy"

_jcm, 2022, doi:10.3390/jcm11175000_

Round 1
Reviewer 1 Report
This article is devoted to actual problem of emergency cardiology, well-written in a common language.
However, there are a number of shortcomings:
1. In my opinion, the title of the article is not correct, because it is likely that the purpose of the scientists was to assess the prognosis or outcomes using MRI, but not only the results of MRI. The title needs to be changed.
2. There is a typo "The effect of AT in the treatment of STEMI is de-batable" (abstract)
3. On what basis was the criterion of adverse LV remodeling of 12% taken? there are many articles where different criteria are indicated, including 15%, 20%. In connection with which it is worth referring to a specific source from which this criterion was taken.
4. In the conclusion the sentence “Considering the small sample size, further studies 356 are required to confirm the place of aspiration thrombectomy in clinical practice” could be omitted, because it is in the limitations of the study.
5. According to the design, 8 people were excluded, and according to the final tables, there are 6 of them. Perhaps this is a typo, or it is worth explaining where these 2 people have gone.
6. In addition, in the introduction it is worth indicating why a more time-consuming and expensive method was used in this study - MRI, and not a fast and inexpensive method of echocardiography.
Author Response
This article is devoted to actual problem of emergency cardiology, well-written in a common language.
Thank you for your opinion and the insightful comments.
However, there are a number of shortcomings:
- In my opinion, the title of the article is not correct, because it is likely that the purpose of the scientists was to assess the prognosis or outcomes using MRI, but not only the results of MRI. The title needs to be changed.
Thank you for raising this issue. We have modified the title, following your suggestion. In the current study we have focused on the CMR outcomes, which may translate to prognosis. Nevertheless, the clinical endpoints were not directly assessed. Therefore we feel it would be an overstatement to include patient prognosis in the title.
- There is a typo "The effect of AT in the treatment of STEMI is de-batable" (abstract)
This has been corrected.
- On what basis was the criterion of adverse LV remodeling of 12% taken? there are many articles where different criteria are indicated, including 15%, 20%. In connection with which it is worth referring to a specific source from which this criterion was taken.
Thank you for pointing this out. We have chosen the definition of a 12% cut-off point for LV remodeling based on a thorough analysis by Bulluck et al., which had been referenced in the ‘Methods’ section concerning CMR assessment (previous reference no. 17, currently no. 15). Following your comment, we have included this reference on the first mention, i.e. in the paragraph concerning LVEDV enlargement (line 106 of the revised manuscript).
- In the conclusion the sentence “Considering the small sample size, further studies are required to confirm the place of aspiration thrombectomy in clinical practice” could be omitted, because it is in the limitations of the study.
Following your suggestion, we have erased the aforementioned sentence from the ‘Conclusions’ section.
- According to the design, 8 people were excluded, and according to the final tables, there are 6 of them. Perhaps this is a typo, or it is worth explaining where these 2 people have gone.
This was a typing error and has been corrected.
- In addition, in the introduction it is worth indicating why a more time-consuming and expensive method was used in this study - MRI, and not a fast and inexpensive method of echocardiography.
Thank you for this comment. We agree that the advantages of CMR over echocardiography have not been emphasized enough. As suggested, we have added arguments for the use of CMR imaging in the assessment of left ventricle parameters post-MI in the revised ‘Introduction’ section.

Reviewer 2 Report
Analysis of the state of the microcirculation perfusion after reperfusion in STEMI is currently an important aspect of developing a treatment strategy for these patients to improve long-term prognosis.
The use of visualization methods for this purpose is a modern and informative method. At present, aspiration thrombectomy is not recommended for routine PCI. All of the above indicates that this study raises an important question regarding the choice of an adequate strategy for managing a patient with STEMI It is very important that the authors were able to recruit two groups of patients that did not differ significantly in age, gender, the presence of risk factors, the absence of severe complications of the disease, coronary anatomy, and the initial degree of myocardial damage. Another advantage of the study is that patients are examined after 6 months of follow-up The article contains tables and figures that well illustrate the obtained results and materials. The authors make a very important conclusion, that the addition of AT to standard primary PCI in STEMI patients with a high thrombus burden results in a significantly smaller degree of MVO, larger reduction of MI size and less LV remodeling at six months following PCI for STEMI, as assessed by CMR. The discussion section presents data from other studies, the results of which are consonant with this work. The results obtained, even on a small sample of patients, contribute to the development of clearer indications for aspiration thrombectomy in patients with STEMI.
It is possible to plan a larger similar study with a large number of patients, using additional laboratory methods that more accurately indicate myocardial damage (troponin, inflammation markers). Such studies may change the level of recommendations for aspiration thrombectomy in patients with STEMI
Author Response
Thank you for your favourable opinion. We agree that a study on a larger number of patients is needed and that an assessment of a broader range of biomarkers could bring important insights into the role of AT in the treatment of STEMI patients with a high thrombus burden. We have added an appropriate comment on this in the revised ‘Study limitations’ section.
Round 2
Reviewer 1 Report
The authors have made all necessary corrections. However, the title of the article confuses me, at the discretion of the authors, I would recommend making it more concise and attractive